# Intracellular antibody signalling is regulated by phosphorylation of the Fc receptor TRIM21

Claire Dickson[†], Adam J Fletcher[†], Marina Vaysburd, Ji-Chun Yang, Donna L Mallery, Jingwei Zeng, Christopher M Johnson, Stephen H McLaughlin, Mark Skehel, Sarah Maslen, James Cruickshank, Nicolas Huguenin-Dezot, Jason W Chin, David Neuhaus, Leo C James*

Medical Research Council Laboratory of Molecular Biology, Cambridge, United Kingdom

**Abstract** Cell surface Fc receptors activate inflammation and are tightly controlled to prevent autoimmunity. Antibodies also simulate potent immune signalling from inside the cell via the cytosolic antibody receptor TRIM21, but how this is regulated is unknown. Here we show that TRIM21 signalling is constitutively repressed by its B-Box domain and activated by phosphorylation. The B-Box occupies an E2 binding site on the catalytic RING domain by mimicking E2-E3 interactions, inhibiting TRIM21 ubiquitination and preventing immune activation. TRIM21 is derepressed by IKKβ and TBK1 phosphorylation of an LxxIS motif in the RING domain, at the interface with the B-Box. Incorporation of phosphoserine or a phosphomimetic within this motif relieves B-Box inhibition, promoting E2 binding, RING catalysis, NF-κB activation and cytokine transcription upon infection with DNA or RNA viruses. These data explain how intracellular antibody signalling is regulated and reveal that the B-Box is a critical regulator of RING E3 ligase activity.

DOI: https://doi.org/10.7554/eLife.32660.001

*For correspondence:
lcj@mrc-lmb.cam.ac.uk

[†]These authors contributed equally to this work

Competing interests: The authors declare that no competing interests exist.

## Introduction

Antibodies are an essential component of protective immunity and their induction is a major aim of vaccination. They mediate a sufficiently rapid inflammatory response upon pathogen re-exposure that infection is halted before host colonization can occur. However, the consequences of inappropriate antibody-induced inflammation are severe, including lethal cytokine storms, autoimmunity and chronic inflammatory disease (*Suntharalingam et al., 2006*). It is therefore essential that antibody immunity is precisely regulated and tightly controlled. Antibody Fc receptors (FcRs) are essential for the diverse effector responses mediated by antibodies (*Woof and Burton, 2004*). Classical FcRs are expressed on the surface of immune cells. Their signalling induces phagocytosis, antigen presentation, antibody dependent cellular cytotoxicity (ADCC), degranulation and pro-inflammatory cytokine production (*Guilliams et al., 2014*). Regulating FcR signalling controls the antibody response. FcR immune signalling is influenced at multiple levels by; varying cell type expression and abundance, balancing the activity of inhibitory vs activatory receptors, requiring receptor crosslinking, setting minimal activation thresholds and by synergising with other pattern recognition receptors (PRRs) (*Vogelpoel et al., 2015*). These diverse mechanisms, deciphered over many decades, operate individually and synergistically to precisely regulate extracellular antibody immunity (*Bruhns and Jönsson, 2015*).

Recent work from our lab has shown that, in addition to classical FcR immunity mediated at the cell-surface, antibodies exert potent immune function from inside cells via a unique cytosolic IgG

**eLife digest** Antibodies are molecules made by the immune system that protect us from infections. They were discovered over 100 years ago, and for most of that time scientists thought they only worked in the bloodstream. Yet recent research showed that when a virus infects our cells it also carries antibodies in with it. Once inside the cell, a protein called TRIM21 recognises the antibody-covered virus and – together with other proteins called ubiquitin enzymes – targets it for destruction via the cell's waste disposal system. At the same time TRIM21 sends a signal to the cell's nucleus to activate certain genes that protect cells across the body from subsequent infection.

The genes activated by TRIM21 have potent antiviral activity. Yet they can also damage the body's own tissues if they are switched on at the wrong time, which may lead to autoimmune diseases like rheumatoid arthritis and multiple sclerosis. It is thus critical that TRIM21 is carefully regulated and only activated during an infection, but it was not clear how this is achieved.

Dickson, Fletcher et al. now show that although TRIM21 is made all the time and is always ready to detect an incoming virus it is made in an inactive state. This is because part of TRIM21, called a B-Box, inhibits the protein's own activity. This was an unexpected discovery because, although the B-Box domain is found in around 100 other human proteins, it was unclear what it did. Dickson, Fletcher et al. then combined structural biology and biochemical approaches to show that the B-Box mimics specific enzymes that work with TRIM21, and blocks them from binding to it. This keeps TRIM21 in an inactive state.

Next, Dickson, Fletcher et al. discovered that TRIM21 becomes active when a chemical tag, specifically a phosphate group, is added to the protein. This modification displaces the B-Box, allowing ubiquitin enzymes to bind to TRIM21 and switch on its activity. Further experiments then showed that this process helps regulate TRIM21 during infections with different viruses, including rhinovirus – the virus behind the common cold – and adenovirus – a common cause of respiratory infection.

Antibodies are now used to treat many medical conditions, but present technologies are based on our understanding of how antibodies work outside cells. By revealing the basis of antibody immunity inside cells, these new findings may lead to new treatments for a range of conditions. Future studies could also explore how failures in the TRIM21 system contribute to autoimmune diseases.

DOI: https://doi.org/10.7554/eLife.32660.002

receptor called TRIM21 (*James, 2014*; *Mallery et al., 2010*; *McEwan et al., 2013*). TRIM21 is structurally, functionally and evolutionarily distinct from surface FcRs and binds antibodies with a significantly higher affinity (*James et al., 2007*). TRIM21 is also distinct in being constitutively and ubiquitously expressed in most cell types and tissues. TRIM21 protects cells from pathogen infection by stimulating a dual effector and sensor response. Upon infection, antibody-coated pathogens are detected in the cytosol by TRIM21. TRIM21 recruits cellular degradation machinery, including the AAA ATPase VCP and the proteasome, resulting in the destruction of viral particles and neutralization of infection. TRIM21 simultaneously activates immune transcription pathways, including NF-κB, leading to potent upregulation of pro-inflammatory cytokines including TNF, CXCL10, IL-6 and IFN (*McEwan et al., 2013*). TRIM21 activation results in cytokine upregulation in mice within hours of infection and is a significant component of the inflammatory response induced by protective antibodies (*Watkinson et al., 2015*). Deletion of TRIM21 compromises the efficiency of protective antibody immunity and causes virus-induced mortality (*Vaysburd et al., 2013*).

In contrast to cell-surface FcRs, and despite its capacity for such rapid and potent inflammatory signalling, it is unclear exactly how TRIM21 is regulated. The regulatory mechanisms that exist for classical FcRs are not applicable to TRIM21. TRIM21 is not expressed at the cell surface and therefore cannot be activated by clustering in lipid rafts or employ this as a mechanism to discriminate monomeric IgG from immune complex. Furthermore, as TRIM21 is dimeric, it is capable of binding both heavy chains of a single IgG simultaneously and forming a 1:1 complex. This is in contrast to other FcRs, which bind asymmetrically to a single heavy chain. There is also no known complementary inhibitory receptor to TRIM21 as exist for surface FcRs. Finally, TRIM21 lacks the ITAM and ITIM

motifs that drives the function of other FcRs, and whose signalling can be precisely regulated by recruitment of Src family kinases and inositol-lipid phosphatases (*Vogelpoel et al., 2015*).

TRIM21 is a member of a large family of ~100 proteins with a common 'tripartite motif' architecture, comprising N-terminal RING, B-Box and coiled-coil domains. Instead of recruiting Src and Syk kinases, TRIM21 stimulates immune signaling via the activity of its RING domain, which catalyses the synthesis of polyubiquitin chains. TRIM21 first recruits the E2 Ube2W to modify itself with an N-terminal monoubiquitin (*Fletcher and James, 2016*). This acts as a primer for the synthesis of an anchored K63-chain catalysed by the heterodimer Ube2N/2V2. TRIM21 is further modified with K48-ubiquitin chains, culminating in proteasomal recruitment. The 19S-resident proteasomal deubiquitinase Poh1 subsequently liberates K63 chains from TRIM21 (*Fletcher and James, 2016*), and these then activate signal transduction pathways via TBK1, TAB/TAK and NEMO. We therefore hypothesized that TRIM21 immune signaling must be regulated at the fundamental level of its ubiquitination activity.

Here we show that initiating ubiquitination is a key step in regulating TRIM21 activity and that a novel mechanism of autoinhibition and kinase-induced E2 enzyme recruitment controls its potent pro-inflammatory activity.

## Results

To investigate the regulation of TRIM21 we tested whether over-expression in the absence of antibody stimuli triggers NFκB. As a positive control, we compared its activity to that of TRIM5α, a related antiviral TRIM protein whose over-expression is sufficient to drive spontaneous NFκB signalling (*Pertel et al., 2011*). While transfection of full-length TRIM5α caused dose-dependent activation of NFκB, full-length TRIM21 failed to activate NFκB at any dose (*Figure 1A*). When activated, both TRIM21 and TRIM5α autoubiquitinate, thereby making themselves a degradative substrate for the proteasome. Consistent with the NFκB result, while endogenous TRIM5α underwent rapid recycling we found little evidence of TRIM21 turnover even after several hours (*Figure 1B* and *Figure 1—figure supplement 1*). Together these observations suggest either that TRIM21 is constitutively inactive and its ubiquitination activity tightly regulated or it is a less active E3 ligase than TRIM5α. However, comparing the ubiquitination activity of TRIM5α and TRIM21 isolated RING domains (T5-R or T21-R) in vitro revealed that TRIM21 was the more active E3, capable of forming ubiquitin chains within minutes (*Figure 1C*). RING dimerization via higher order assembly is the proposed mechanism by which TRIMs autoregulate (*Wagner et al., 2016*). TRIM5α self-assembles and mutations that disrupt RING dimerization inhibit catalytic ability (*Yudina et al., 2015*). We therefore introduced structurally corresponding mutations into TRIM21 (*Figure 1D* and *Figure 1—figure supplement 2*) to test whether RING dimerisation is required for activity. SEC MALS and AUC demonstrated that while the wild-type RING is in monomer-dimer equilibrium, mutants M10E and M10E/M72E do not undergo detectable dimerization (*Figure 1E and F*). However, both mutants remained active and could catalyse unanchored K63 chain synthesis, Ube2W-primed anchored K63 chain extension and ubiquitin discharge (*Figure 1G–I*), albeit less efficiently than wild-type. These results demonstrate that a preformed TRIM21 RING dimer is not a pre-requisite for ubiquitination activity.

RING dimerization is not required for E2 enzyme binding, as all interactions occur within a single monomer (*Yudina et al., 2015*). Instead, dimerization of RING E3s is thought to be needed for additional contacts with the E2-charged ubiquitin. Mutations in the partner RING at this interface reduce TRIM25 catalytic activity in vitro (*Figure 2A*)(*Sanchez et al., 2016*). Equivalent mutations in TRIM21 had no impact on ubiquitination, confirming that RING dimerization is not an intrinsic requirement for TRIM21 activity (*Figure 2B–D*). The mutation E10R reduces ubiquitination by TRIM25, possibly by disrupting a hydrogen bond with ubiquitin (*Koliopoulos et al., 2016*). We hypothesised that the corresponding glutamate (E13) in TRIM21 may allow a monomeric RING to contact both E2 and bound ubiquitin simultaneously, obviating the requirement for dimerization. Consistent with this hypothesis, E13R reduced TRIM21 catalysis (*Figure 2D*). Taken together these data show that the monomeric TRIM21 RING is a highly active E3. Thus, the lack of constitutive TRIM21 NFκB activity must be the result of tight regulation, independent of higher order oligomerisation.

The next domain in the tripartite motif of TRIM21 after the RING is the B-Box2 (hereafter referred to as 'B-Box'), a domain largely unique to TRIM proteins but whose function (besides an ability to oligomerise in TRIM5α) is unknown. Comparing the activities of RING, RING-Box and MiniTRIM21

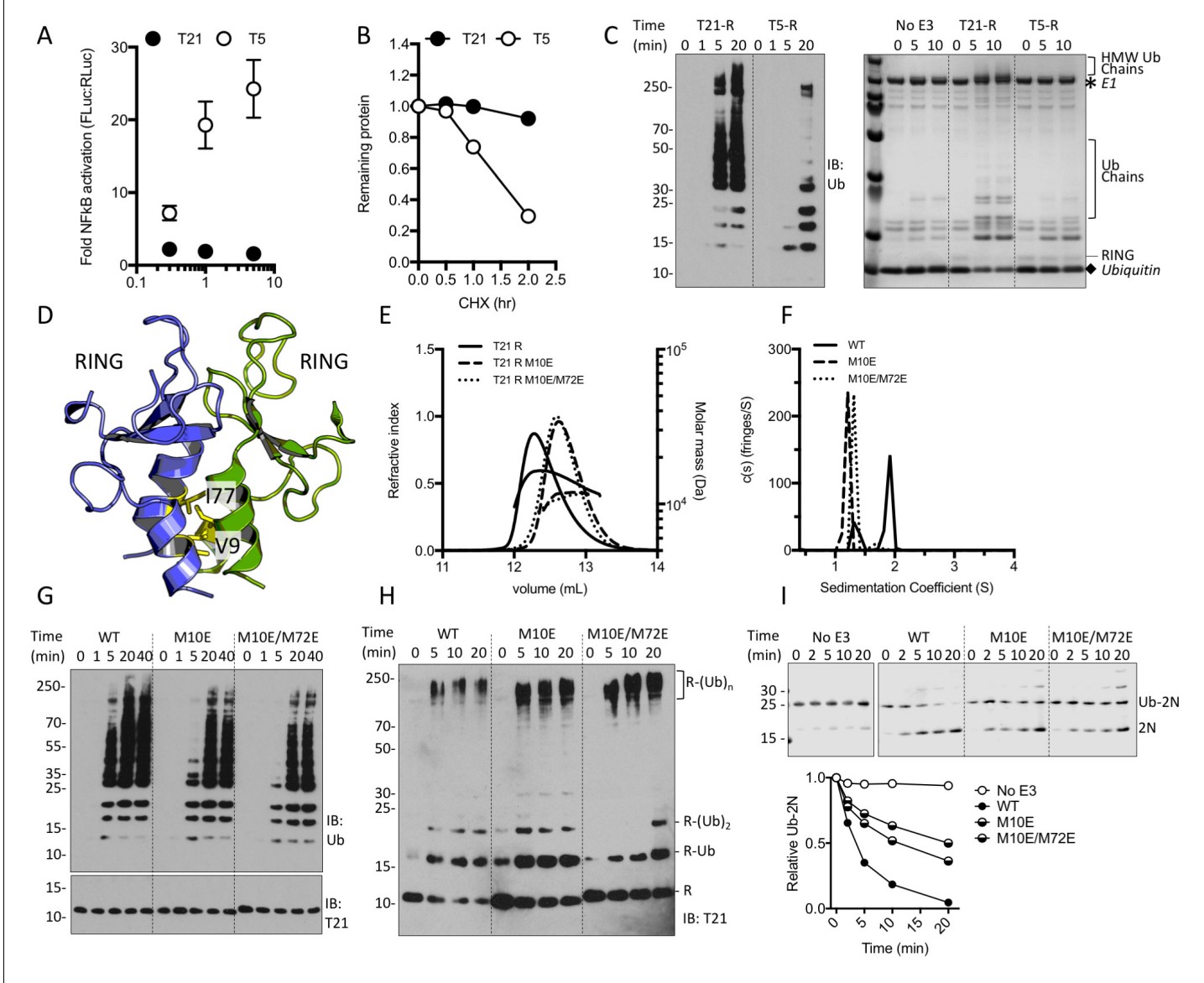

**Figure 1.** TRIM21 NFκB signalling is constitutively silent despite monomeric RING K63 chain synthesis. (**A**) TRIM5α (T5) but not TRIM21 (T21) overexpression in 293Ts activates an NFκB-luciferase reporter. (**B**) Reduction in endogenous T5 and T21 protein levels after cycloheximide treatment. (**C**) (left) Immunoblot for in vitro polyubiquitin synthesis catalysed by T5 or T21 RING (**R**) domains in the presence of Ube2N/Ube2V2; (right) SDS-PAGE showing T21 R autoubiquitination in the presence of Ube2W and Ube2N/Ube2V2. (**D**) RING dimerization interface in T5 structure (4TKP). (**E–F**) Oligomerisation of T21 R and mutants M10E and M10E/M72E as assessed by (**E**) SEC MALS and (**F**) AUC. (**G–H**) Wild-type (WT) T21 R or dimerization mutants catalyzing unanchored polyubiquitin synthesis using Ube2N (**G**) or anchored chains using Ube2N/Ube2V2 and Ube2W (**H**). (**I**) T21 (WT or mutants) catalysed discharge of ubiquitin from conjugated Ube2N/Ube2V2.

DOI: https://doi.org/10.7554/eLife.32660.003

The following figure supplements are available for figure 1:

**Figure supplement 1.** Cellular Turnover of endogenous TRIM5α and TRIM21.

DOI: https://doi.org/10.7554/eLife.32660.004

**Figure supplement 2.** Sequence alignment of TRIM5 and TRIM21.

DOI: https://doi.org/10.7554/eLife.32660.005

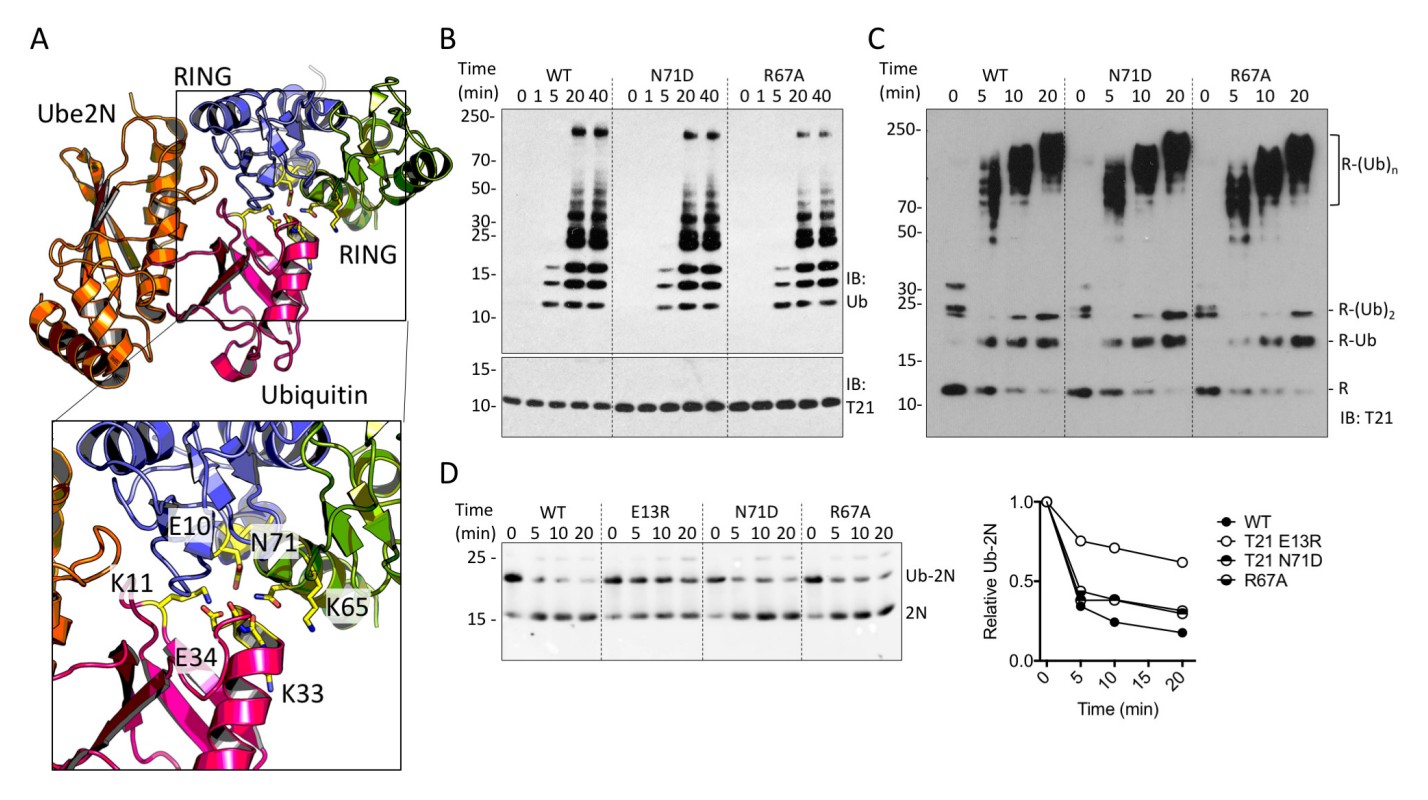

**Figure 2.** TRIM21 RING dimerization is not required to catalyse ubiquitin release from E2:Ub. (A) Model of the TRIM25 RING:Ube2N:Ub complex (5EYA), showing interactions between ubiquitin and both RING monomers. (B–D) Ubiquitination activity of WT T21R and mutants of putative ubiquitin-contacting residues in the second monomer. (B) Catalysis of unanchored ubiquitin chains using Ube2N. (C) Synthesis of anchored K63-chains on TRIM21 in presence of both Ube2N/Ube2V2 and Ube2W. (D) Catalysis of ubiquitin discharge from conjugated Ube2N/Ube2V2.
DOI: https://doi.org/10.7554/eLife.32660.006

proteins (based on a monovalent MiniTRIM5 that preserves the Box-coiled coil interface (*Wagner et al., 2016*)), revealed that the presence of the B-Box significantly reduces ubiquitination (*Figure 3A*). This is due to direct inhibition of catalytic activity, as the B-Box prevented ubiquitin discharge from either Ube2D1 or Ube2N (*Figure 3B*). To understand how the B-Box modulates TRIM21 catalysis we solved a 2 Å crystal structure of the RING-Box protein (*Supplementary file 1*). Surprisingly, this revealed that the B-Box occupies the E2 binding site on the RING domain (*Figure 3C*). Comparison to the TRIM5α RING:E2 complex shows that the B-Box is located where helix 1 of the E2 would normally be positioned during RING binding (*Figure 3D*). Crucially, the B-Box is able to occupy the E2 binding site on the RING by acting as an E2 mimic, forming similar electrostatic interactions. The RING surface contains a negatively charged patch, which is complementary to a strongly positively charged patch on both the E2 and the B-Box (*Figure 3E*). In particular, Box-RING interaction is stabilized through a salt bridge between residues R118 and E12, which is analogous to the interaction between E2 residue R14 and RING residue E11 in the TRIM5:Ube2N structure (*Figure 3D*). To test directly whether the B-Box prevents E2 binding we assigned $^{15}$N HSQC spectra of monomeric (M10E) RING and RING-Box constructs and measured chemical shift perturbations (CSPs) upon titration with Ube2N (*Figure 4* and *Figure 4—figure supplement 1*). Titration of Ube2N into the RING resulted in significant CSPs in E2 interface residues (denoted by the green shaded regions in *Figure 4A*). In contrast, no evidence of Ube2N binding to the RING-Box was observed. Taken together, these data suggest that the B-Box may be an autoinhibitory domain whose function is to regulate RING activity by preventing E2 recruitment.

Next, we investigated what relieves B-Box inhibition and promotes TRIM21 ubiquitination activity. Recently it was shown that unrelated innate immune adaptors including MAVS, STING and TRIF all contain a short *pLxIS* motif (where *p* is hydrophilic, *x* is non-aromatic and S is the target serine),

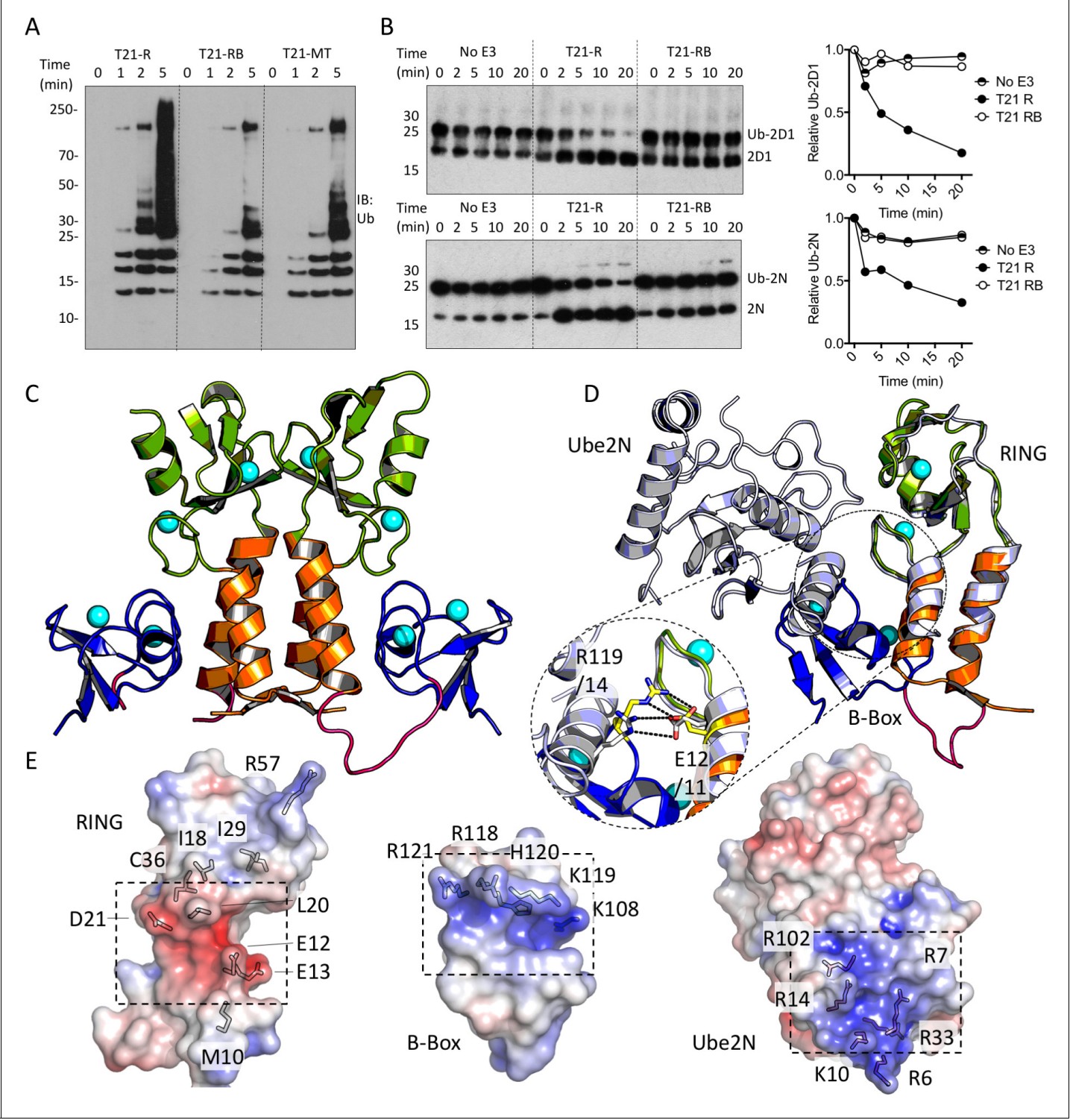

**Figure 3.** The Box is an autoinhibitory domain which supresses E3 ligase activity. (A) Catalysis of Ube2N/Ube2V2 driven polyubiquitin-chain synthesis by T21 RING (R), RING-Box (RB) and MiniTRIM21 (MT). (B) T21 R or RB catalysed ubiquitin discharge from Ube2D1 or Ube2N. (C) 2 Å X-ray structure of T21 RB coloured by domain with zinc ions in cyan. (D) Superposition of T21 with T5 model 4TKP (grey), showing Ube2N and the Box are structurally competitive. (E) Surface representation of (left-to-right) T21 R, Box and Ube2N binding sites coloured by electrostatic potential from −5 kT/e (red) to + 5 kT/e (blue).

DOI: https://doi.org/10.7554/eLife.32660.007

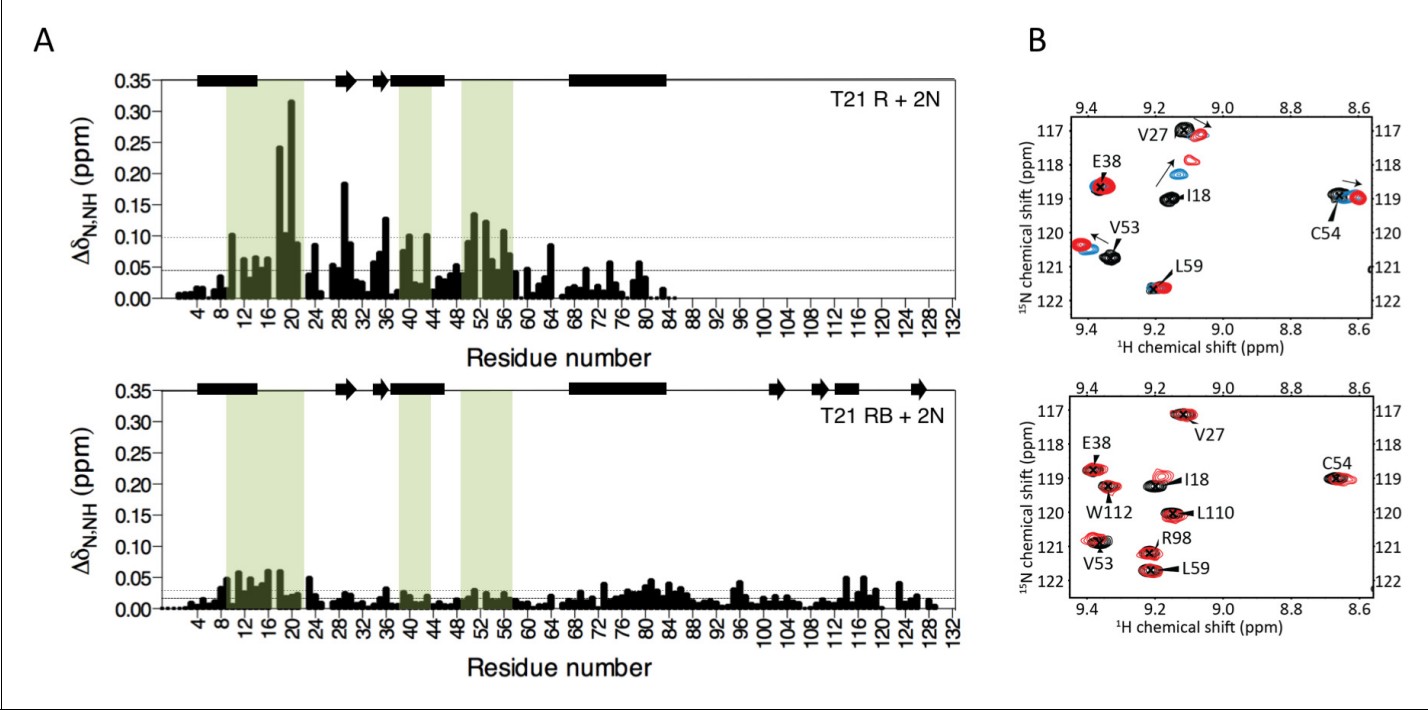

**Figure 4.** The Box prevents binding of E2 enzyme to the RING E3. (**A**) Chemical shift perturbations recorded upon addition of unlabelled Ube2N into $^{15}$N labelled TRIM21 RING (R) (top) or RING-Box (RB) (bottom). E2 interacting regions are highlighted in green and secondary structure elements comprising either helices (rectangles) or b-strands (arrows) are indicated at the top of each plot. (**B**) Examples of the $^{15}$N HSQC spectra; residues in the RING (top) are shifted upon adding E2 (RING:E2 ratio: black 1:0, blue, 1:0.5, red, 1:1), while the same residues in the RING-Box (bottom) show little or no movement at a 1:1 titration point (red). *Figure 4—figure supplement 1* shows the complete spectra.

DOI: https://doi.org/10.7554/eLife.32660.008

The following figure supplement is available for figure 4:

**Figure supplement 1.** HSQC of TRIM21 RING and RING-Box with Ube2N.
DOI: https://doi.org/10.7554/eLife.32660.009

which recruits kinases IKKβ or TBK1 resulting in serine phosphorylation and signal potentiation (*Liu et al., 2015*). Remarkably, TRIM21 contains a very similar motif at the end of its RING domain. Moreover, the target serine in this motif is located at the centre of the B-Box:RING interface (*Figure 5A*) and undergoes phosphorylation in cells (*Figure 5B*). We hypothesized that this could provide a mechanism to activate TRIM21 during an immune response. To investigate this further, we raised specific antisera against the phosphoserine peptide $^{67}$RQLANMVNNLKEISQ$^{81}$ (*Figure 5C*). Using this anti-pS80 serum, we detected cellular phosphorylation of TRIM21 but not an S80A mutant, upon IKKβ overexpression (*Figure 5D*). To ensure that IKKβ was acting directly rather than inducing the expression of a second kinase via NFκB, we repeated the experiment in the presence of proteasome inhibitor Bortezomib, which inhibits the degradation of IκBα and blocks NFκB activation. Efficient IKKβ phosphorylation of TRIM21 was observed even under conditions where NFκB signaling was abolished (*Figure 5E*). To confirm that IKKβ directly phosphorylates TRIM21, we incubated recombinant RING-Box and IKKβ proteins in vitro, observing ATP-dependent IKKβ phosphorylation of TRIM21 that was lost upon mutation of motif residues or addition of an IKKβ inhibitor (*Figure 5F–H*). Robust IKKβ-mediated phosphorylation was also observed using full-length TRIM21 protein (*Figure 5I*). Finally, knock-out of IKKβ by CRISPR/Cas9 abolished modification of overexpressed TRIM21-His (*Figure 5J*), suggesting that endogenous kinase phosphorylates TRIM21.

Previously, it has been shown that while MAVS is phosphorylated by IKKβ and TBK1, STING and TRIF are only phosphorylated by TBK1 suggesting that there is differential kinase dependence even though signaling adapters share the consensus *p*LxIS motif (*Liu et al., 2015*). To test whether TRIM21 is a substrate for TBK1 phosphorylation, we incubated the recombinant full-length protein with the kinase in the presence of ATP. TBK1 was capable of phosphorylating TRIM21 in vitro

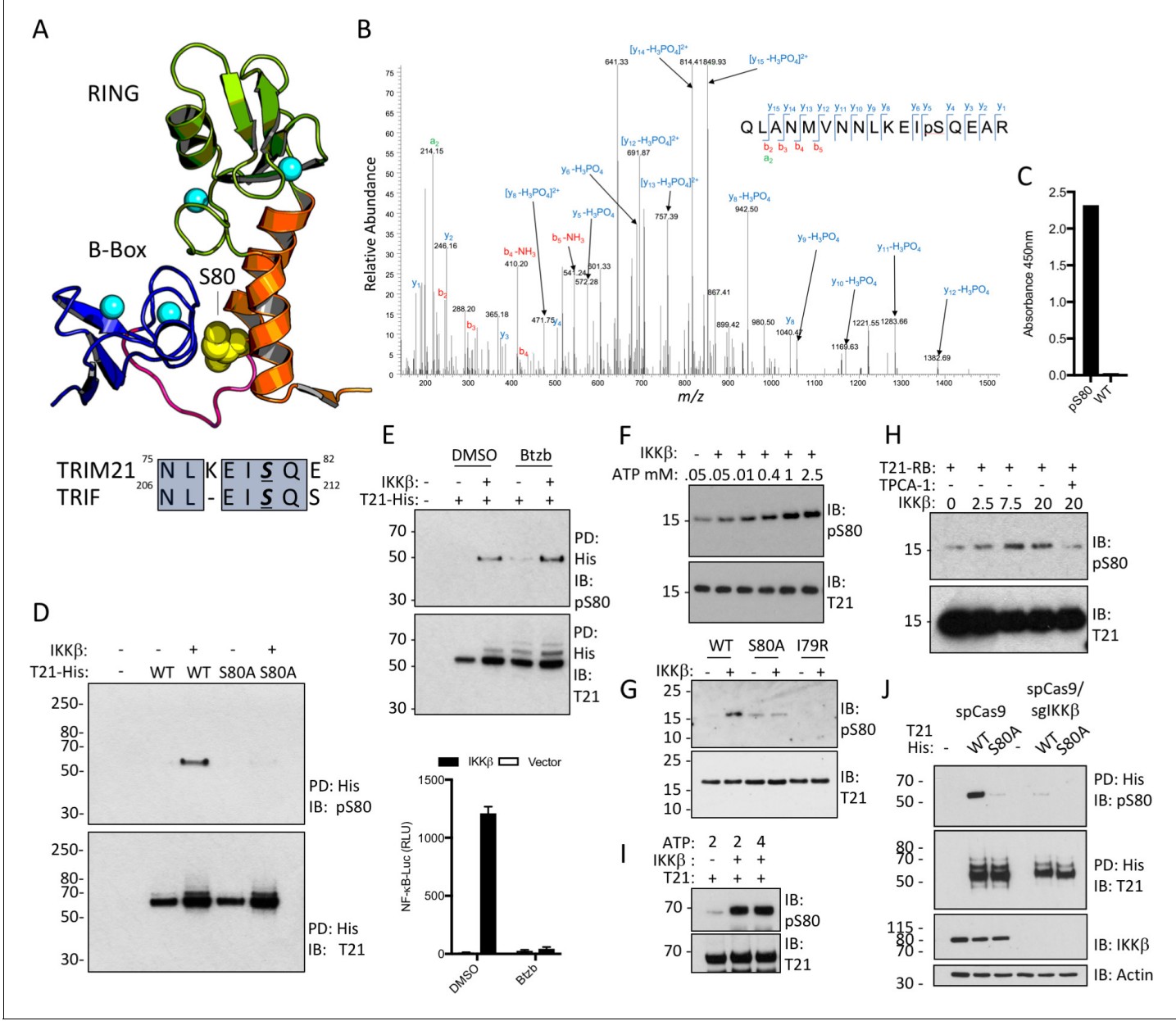

**Figure 5.** TRIM21 possesses a *p*LxxIS motif in its RING domain and is phosphorylated in vitro and in cells. (**A**) Residue S80 (yellow spheres) is located at the centre of the Box:RING interface and is part of an epitope related to the IKKβ phosphorylation motif found in TRIF. (**B**) MS/MS spectra of TRIM21-His purified from 293 T cells indicating the presence of phosphorylated S80 (pS80) within a tryptic peptide precursor/m/z/680.03 ± 0.39 ppm. (**C**) ELISA of pS80 antisera at 1:50000 against recombinant WT or pS80 TRIM21. (**D**) IKKβ phosphorylates C-terminally His-tagged WT but not S80A T21-His, as detected by specific pS80 polyclonal sera. (**E**) As D except in presence of 30 nM bortezomib (Btzb) or DMSO and, below, NFκB stimulation by IKKβ in parallel. (**F–H**) In vitro phosphorylation of T21 RB by IKKβ with increasing ATP (**F**), impaired phosphorylation of S80A and I79R RB (**G**), in presence of 20 μM IKKβ inhibitor TPCA-1 (**H**). (**I**) In vitro IKKβ phosphorylation of full-length lipoyl-T21. (**J**) Phosphorylation of WT or S80A T21-His from control 293T (spCas9) or IKKβ-deleted 293T (spCas9/sgIKKβ) cells.

DOI: https://doi.org/10.7554/eLife.32660.010

comparably with IKKβ (*Figure 6A*). Overexpressed TBK1 was also capable of strongly phosphorylating cellular TRIM21 and causing significant stabilization of the protein (*Figure 6B*). To overcome limitations of kinase overexpression we sought to stimulate TRIM21 phosphorylation by activing the upstream pathway. First, we stimulated the TBK1 pathway by expressing one of its upstream adaptors, MAVS. MAVS expression significantly increased phosphorylation of TRIM21 (*Figure 6C*).

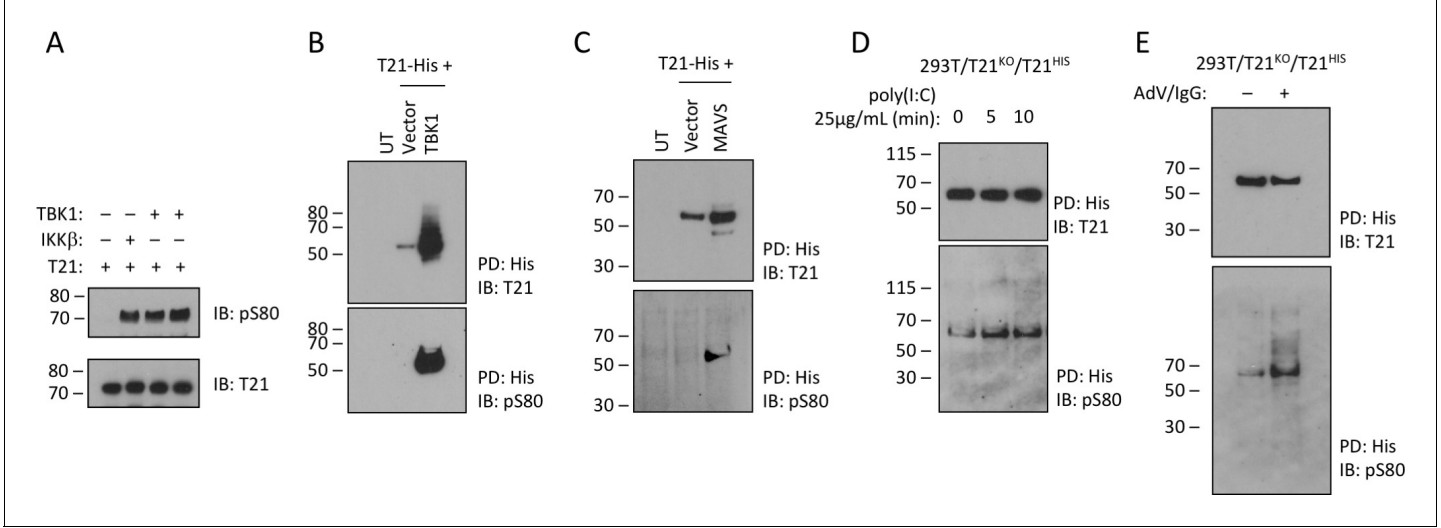

**Figure 6.** TRIM21 is phosphorylated upon immune stimulation. (A) Full-length recombinant lipoyl-T21 protein phosphorylation in vitro by IKKβ and TBK1. (B–C) Expression of TBK1 (B) or upstream adaptor MAVS (C) in 293 T cells phosphorylates and stabilises C-terminally His-tagged T21. (D) 293 T cells lacking T21 reconstituted with His-tagged T21 and challenged with poly(I:C). Increased phosphorylation of T21 detected by immunoblot with pS80 antisera. (E) As in (D), except cells infected with adenovirus 5 (AdV) in the presence of IgG and T21 immunoprecipitated 7 hr post-infection.
DOI: https://doi.org/10.7554/eLife.32660.011

Second, we stimulated cells with the MDA5/RIG-I ligand poly(I:C) (*Kato et al., 2008*). We observed a rapid increase in TRIM21 phosphorylation upon poly(I:C) stimulation, consistent with TBK1 activation (*Figure 6D*). Finally, we monitored changes in TRIM21 phosphorylation following infection with antibody-coated adenovirus, under conditions where we have previously demonstrated TRIM21-dependent antiviral activity (*McEwan et al., 2013*). The data show that in equivalent levels of immunoprecipitated TRIM21 there is a significant increase in the proportion of phosphorylated TRIM21 (*Figure 6E*).

We next investigated whether serine phosphorylation can act as a switch to relieve B-Box inhibition and activate TRIM21. To test this, we introduced the phosphomimetic mutation S80E into the RING-Box and measured its ability to catalyse ubiquitin discharge. Remarkably, introduction of S80E into the RING-Box was sufficient to completely restore catalytic activity to the same level as the RING alone (*Figure 7A–B*). While glutamate is a routinely used phosphomimetic it does not fully recapitulate a phosphoserine. We therefore utilized an evolved orthogonal aminoacyl-tRNA synthetase/tRNA$_{CUA}$ pair (*Rogerson et al., 2015*) to incorporate phosphoserine co-translationally at position 80 in MiniTRIM21 protein. Comparison with wild-type protein revealed that S80 phosphorylation potentiates TRIM21 ubiquitination activity (*Figure 7C*). Phosphorylation-dependence was confirmed by treating the phosphoprotein with phosphatase, which dephosphorylated S80 and restored autoinhibition (*Figure 7C*). Taken together, this show that modification of S80 is sufficient to control ubiquitination activity of TRIM21 in the presence of the B-Box. Moreover, mutant S80E can be used as a functional mimic of serine phosphorylation at position 80.

The above data suggest that TRIM21 is kept in a constitutively inactive state by the B-Box domain, which prevents E2 binding and whose autoinhibition is released upon phosphorylation. To test this, we compared NFκB activation by TRIM21 mutants S80E or S80A with TRIM5α. Neither S80A nor wild-type TRIM21 triggered NFκB, but S80E conferred robust activation (*Figure 8A*). In contrast, mutation of residue S80 did not alter virus neutralization (*Figure 8B*). This is consistent with previous data that TRIM21 signalling has an activation threshold whereas neutralization does not (*Foss et al., 2016*). To demonstrate the importance of TRIM21 phosphorylation in immune sensing during infection, we challenged cells expressing different TRIM21 mutants with human adenovirus (Adv) ± human serum IgG and measured *TNFA* transcription after 4 hr. Significant *TNFA* induction was only observed during infection in the presence of antibody, and not in TRIM21 knockout cells (K21, *Figure 8C* and *Figure 8—figure supplement 1*). Normal *TNFA* induction was

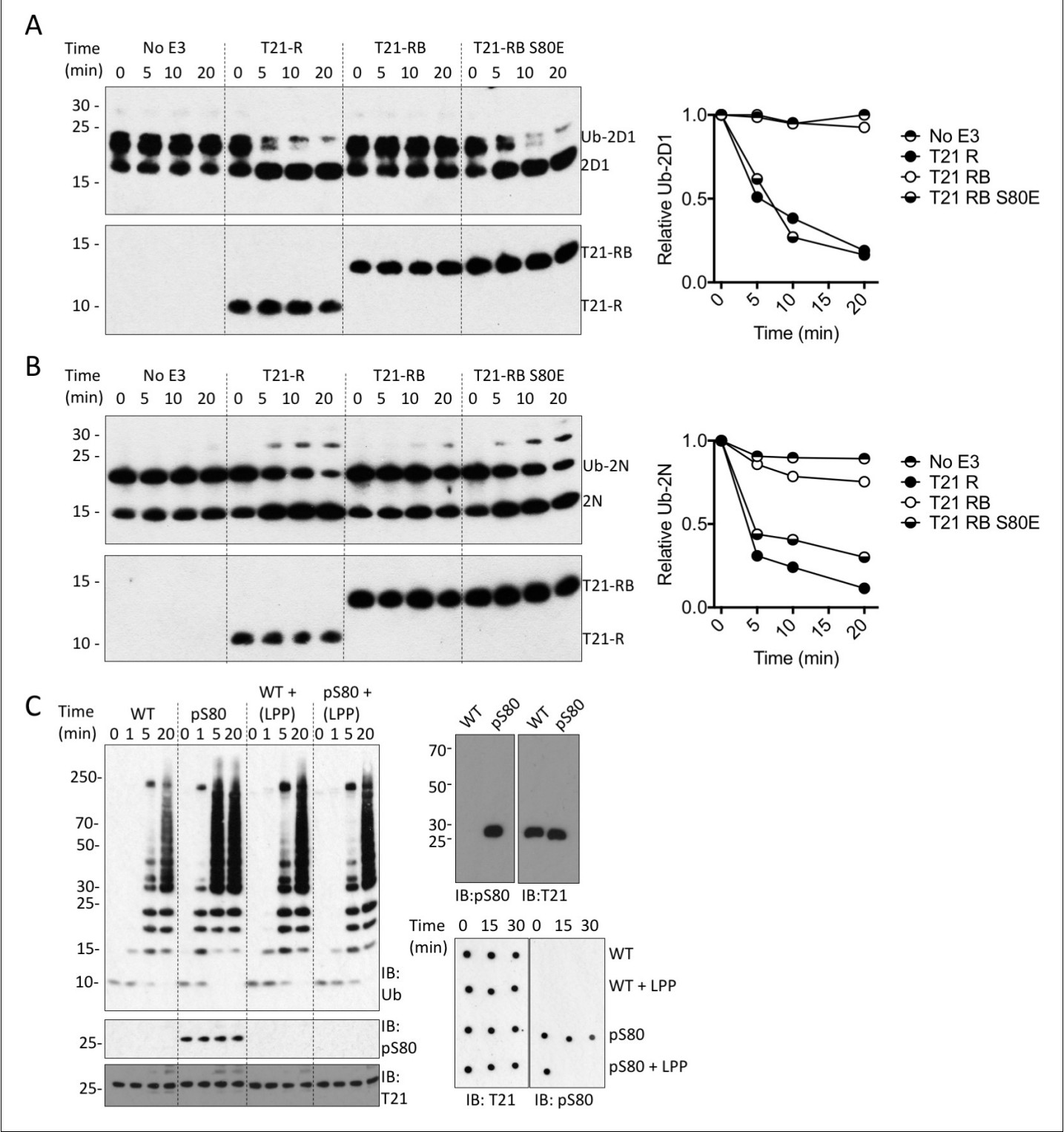

**Figure 7.** Phosphorylation of S80 or phosphomimetic mutation S80E releases Box inhibition and promotes TRIM21 ubiquitination activity. (A–B) Catalysis of ubiquitin discharge from Ube2D1 (A) or Ube2N (B) by TRIM21 RING (R), RING-Box (RB) or RING-Box containing S80E mutation. The phosphomimetic mutation is sufficient to confer RING-like discharge kinetics on the RING-Box (offset graphs). (C) (top) Ubiquitin-chain catalysis by MiniTRIM21 either WT or with a phosphoserine incorporated co-translationally at position 80 by amber suppression (pS80). WT and pS80 MiniTRIM21 blotted with pS80 or T21 sera (top-right) or after treatment with lambda protein phosphatase (LPP) (bottom-right). LPP dephosphorylates S80 and restores Box autoinhibition.

DOI: https://doi.org/10.7554/eLife.32660.012

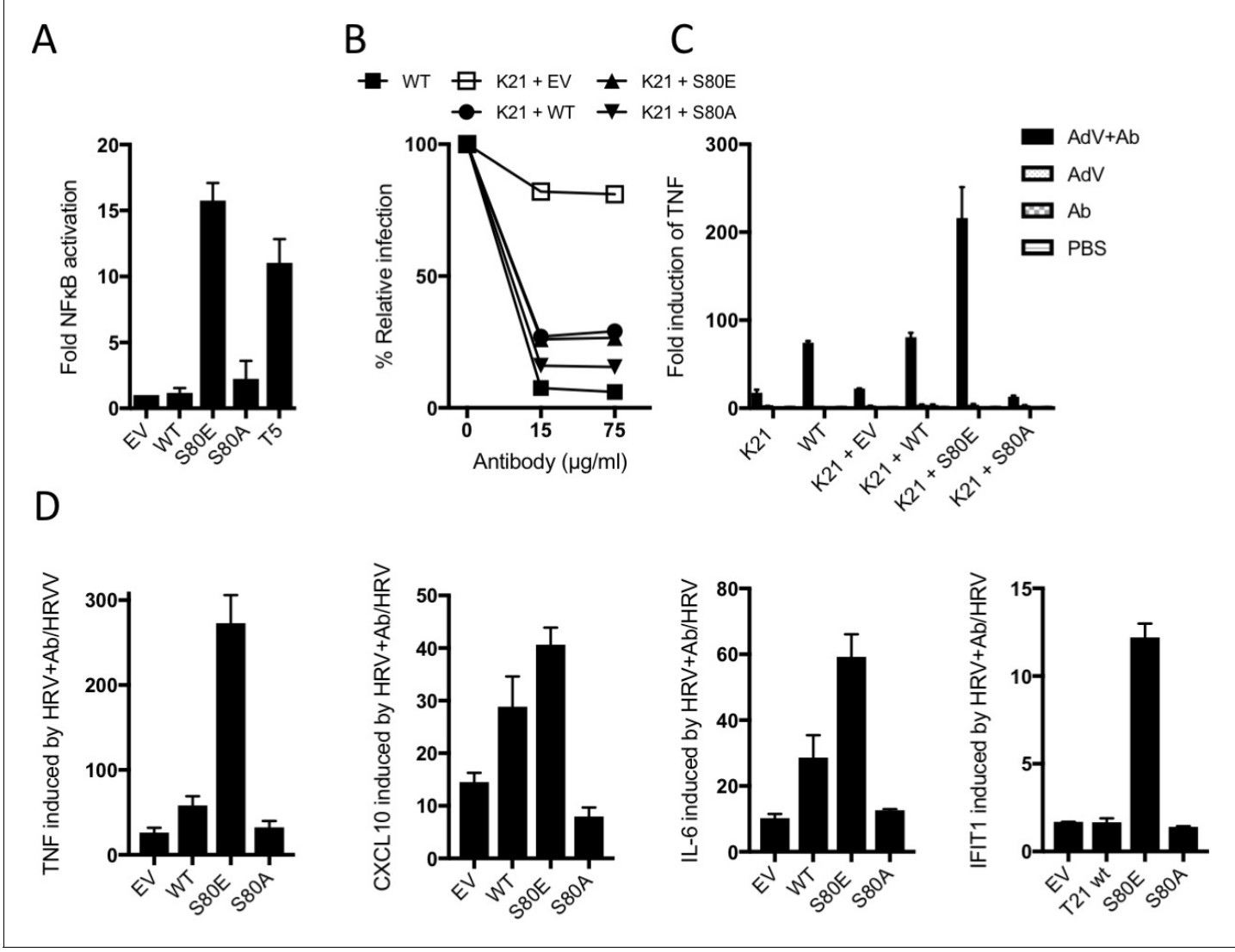

**Figure 8.** B-Box autoinhibition regulates antibody-dependent immune signalling by TRIM21. (**A**) NFκB activation in 293Ts expressing WT, S80A or S80E T21 or WT T5. Phosphomimetic mutation S80E is sufficient to render TRIM21 constitutively active for NFκB activity. (**B**) Adenoviral neutralization is reconstituted in T21 knockout MEFs (K21) equally upon overexpression with WT or S80 mutants. (**C**) T21-mediated *TNFA* induction upon adenoviral (AdV) infection in the presence of antibody (Ab) is restored in K21 by overexpression of WT TRIM21 but not S80A, while S80E is hyperactive. Fold change in *TNFA* transcripts as determined by qPCR. (**D**) Immune gene transcription induced by antibody-coated human rhinovirus (HRV) in reconstituted MEFs. Phosphomimetic mutation S80E potentiates transcriptional activation by TRIM21, while S80A inhibits activity.
DOI: https://doi.org/10.7554/eLife.32660.013

The following figure supplement is available for figure 8:

**Figure supplement 1.** Ectopic expression of TRIM21 in knockout MEFs.
DOI: https://doi.org/10.7554/eLife.32660.014

restored in K21 cells by TRIM21 overexpression, confirming TRIM21-dependence. Importantly, expression of S80A failed to rescue *TNFA* induction, while S80E enabled *TNFA* induction beyond wild-type levels. A change in TRIM21 phosphorylation upon infection could not be detected but this may be because only a fraction of cellular TRIM21 is recruited and modified during the response. This would be consistent with previous data showing that TRIM21 ubiquitination and degradation is also undetectable, despite being required for activity(*Mallery et al., 2010*). To confirm the importance of S80 and B-Box inhibition in regulating TRIM21 immune signaling, we repeated our infection experiments using an unrelated RNA virus, human rhinovirus 14 (HRV). As with AdV, expression of wild-type TRIM21 in knockout cells increased *TNFA* induction upon infection with HRV + antibody

(*Figure 8D*). Mutant S80E greatly potentiated *TNFA* transcription while S80A failed to reconstitute TRIM21 activity. Similar results were obtained for cytokines *CXCL10* and *IL6* and the interferon-stimulated gene *IFIT1*.

To obtain direct evidence that S80 phosphorylation activates TRIM21 by displacing the B-Box domain, we compared the E2 binding of wild-type and S80E RING-Box proteins by NMR. In contrast to wild-type, we detected significant CSPs in canonical E2 binding site residues upon titration of Ube2N into S80E (*Figure 9A*). There were also changes in residues at the base of helix 2, within the linker and on B-Box residues that contact the RING. This is consistent with restoration of E2 binding and the stabilization of an active conformation in which the B-Box domain is displaced. To further demonstrate B-Box displacement, we compared the dynamics of wild-type and S80E proteins. The {$^1$H}$^{15}$N NOE ratios reveal that the most dynamic region is the linker (residues 83–102) between the RING and B-Box (*Figure 9B*). This is in agreement with the higher temperature factors observed for these residues in the crystal structure (*Figure 9—figure supplement 1*). Importantly, mutant S80E has markedly lower enhancement factors both in the linker region and in helix 1 (residues 1–17), against which the B-Box packs (*Figure 9B*). This increase in local motion is consistent with reduced interaction between RING and B-Box in the S80E mutant. To demonstrate that B-Box displacement promotes access to the E2 binding site on the RING, we carried out paramagnetic relaxation enhancement experiments (solvent PREs). In these experiments, addition of the paramagnetic metal

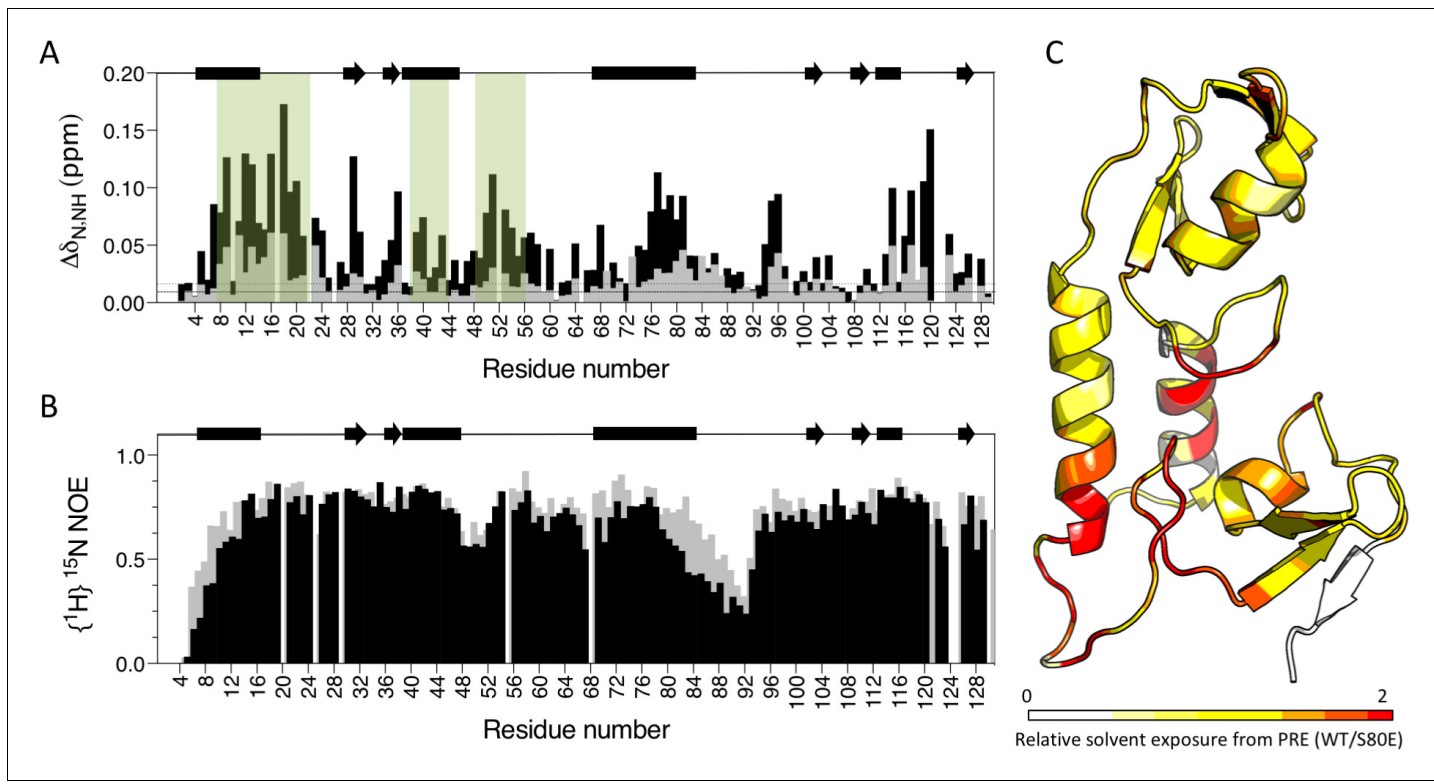

**Figure 9.** Phosphomimetic mutation S80E displaces the Box from the RING and allows E2 enzyme binding. (A) Comparison of chemical shift perturbations upon E2 enzyme Ube2N titration into T21 RB (grey) or T21 RB S80E (black). E2 interacting regions are highlighted in green and secondary structure is indicated by rectangles (helices) and arrows (β-strands). (B) Comparison of heteronuclear NOEs for T21 RB (grey) or T21 RB S80E (black) in absence of E2. The data is consistent with increased dynamics in the linker region between the RING and the Box upon introduction of S80E. (C) Relative solvent exposure in S80E vs. WT, as detected by solvent PREs (shown as intensity ratio WT/S80E mapped onto the RB structure; primary data in *Supplementary file 2*). White areas indicate regions more solvent-exposed in WT while red areas indicate parts of the surface more solvent-exposed in S80E.

DOI: https://doi.org/10.7554/eLife.32660.015

The following figure supplement is available for figure 9:

**Figure supplement 1.** Main chain temperature factors for T21 RING-Box structure.

DOI: https://doi.org/10.7554/eLife.32660.016

complex Gd(DTPA–BMA) selectively attenuates signals from solvent-exposed surfaces (*Pintacuda and Otting, 2002*). The S80E mutation caused a dramatic change in surface accessibility precisely localized to the RING-Box interface (*Figure 9C* and *Supplementary file 2*). These data confirm that phosphomimetic mutation at serine 80 disrupts the RING:Box interaction and uncovers the E2 binding site.

## Discussion

Our results suggest intracellular antibody immune signalling is kept in check via tight regulation of the cytosolic antibody receptor TRIM21. TRIM21 activates immune transduction pathways upon sensing an antibody-coated pathogen by synthesizing K63-ubiquitin chains. Controlling synthesis of these chains allows TRIM21 signalling to be regulated at the fundamental level. Here we have shown that this is accomplished by keeping TRIM21 in a constitutively silent state through an autoinhibitory mechanism that blocks E2 enzyme recruitment. Autoinhibition is mediated by the B-Box, providing a function for this ubiquitous but hitherto enigmatic TRIM protein domain. The B-Box inhibits E2 recruitment by acting as an E2 mimic and competing for RING binding. We also show that TRIM21 contains a variant of an IKKβ and TBK1 phosphorylation motif and is phosphorylated at residue S80. Phosphorylation of S80 or its replacement by a phosphomimetic displaces the B-Box, allowing E2 recruitment and potentiating TRIM21 ubiquitination activity, NF-κB signalling and cytokine transcription upon infection with DNA or RNA viruses.

Although TRIM21 regulation is seemingly distinct from that of cell-surface FcRs, parallels exist. For instance, signalling by both types of antibody receptor involves regulation by kinases. In the case of TRIM21, this involves a $p$LxxIS motif (where $p$ is hydrophilic) and the kinases IKKβ and TBK1 while in other FcRs this involves an YxxL/I motif and recruitment of Src kinases. Importantly, it is not clear for either receptor type exactly how kinase activation occurs. For surface FcRs, this is thought to involve receptor clustering via cross-linking but why this causes Src recruitment and activation is unclear (*Huang et al., 1992*). TRIM21 may also undergo clustering, via cross-linking or higher order assembly, when it is recruited to opsonized pathogens. This may provide a basal activity that is amplified by kinase phosphorylation. The co-option of IKKβ and TBK1 as amplifying kinases suggests that TRIM21 could participate in its own positive-feedback loop. Again there are parallels to surface FcRs, where tyrosine phosphorylation has been shown to promote receptor clustering, which promotes phosphorylation and so on (*Sobota et al., 2005*).

Both types of Fc receptor also synergise with other PRRs and immune signalling pathways to amplify and diversify the resulting inflammatory response. Surface FcRs are known to exhibit crosstalk with Toll-like receptors (TLRs), greatly amplifying the production of cytokines such as TNFα during phagocytosis of opsonized bacteria (*den Dunnen et al., 2012*; *Vogelpoel et al., 2014*). Similarly, TRIM21 has been shown to synergise with cytosolic nucleic acid receptors such as RIG-I and cGAS, resulting in pathogen-specific responses and multiple signalling waves (*Watkinson et al., 2015*). Whether there is direct collaboration between TRIM21 and surface FcRs is currently unknown and will be important to determine in future work. Both immune signalling and antigen processing functions of TRIM21 are well placed to synergise with FcR immune complex capture and sorting activities, for instance to promote class I presentation. Precedence exists in the form of collaboration between FcRn and surface FcRs to drive antigen processing by dendritic cells (DCs)(*Baker et al., 2011*; *Qiao et al., 2008*).

TRIM21 belongs to a large family of structurally related proteins, many of which have known roles in cell signalling. They include other proteins with roles in immune signalling, such as the antiretroviral TRIM5(*14*) and RIG-I activator TRIM25 (*Gack et al., 2007*). While progress has been made in understanding the structure and function of TRIMs, it is still unclear how TRIM proteins are activated and regulated. Existing models of activation were solely based on substrate-induced aggregation or higher order assembly. This was thought to be necessary for dimerization of the RING domains located at either end of the molecule, which was assumed to be a pre-requisite for ubiquitination activity (*Koliopoulos et al., 2016*). However, our data show that this is not the case for TRIM21, and so cannot be assumed for other members of the TRIM family. Furthermore, higher order assembly alone does not explain the discrepancy between in vitro and cellular TRIM ubiquitination activity. TRIM RINGs, including those of TRIM5, 25 and 32, are active in vitro at concentrations where they are monomeric (*Yudina et al., 2015*; *Sanchez et al., 2016*; *Koliopoulos et al., 2016*). Given that

TRIMs are not constitutively active in cells, there must be another level of regulation. It has been proposed that binding of a ubiquitin-conjugated E2 may drive RING equilibrium towards the dimeric form, explaining the in vitro activity. However, this would not explain the absence of activity in cells where the same phenomenon would occur. Moreover, if E2-Ub can drive TRIM RING dimerization then a pre-existing dimer is by definition no longer a pre-requisite for function and higher order assembly cannot be the sole regulator. It seems likely that the activation and regulation of TRIMs is a complex process that is regulated at multiple levels. This is exemplified by TRIM21 whose neutralization and signalling functions are both mediated by ubiquitination, but only signalling requires release of B-Box inhibition. Previously we have observed that TRIM21 signalling has a distinct threshold whereas neutralization takes place at very low antibody occupancy (<2 antibodies) (*McEwan et al., 2012*) and under conditions of sub-optimal TRIM21:antibody and antibody:antigen kinetics (*Foss et al., 2016*; *Bottermann et al., 2016*). Therefore, basal activity via higher order assembly may be sufficient for neutralization but not signalling. Strict regulation of signalling is intuitive, as it ensures a proportional response and avoids inappropriate triggering. Setting activation thresholds and regulating signalling may mitigate against any endosomal leakage of antibodies into the cytosol. The activation of TRIM21 by leaked immune complexes in lysosome-maturation-defective macrophages of lupus-prone mice illustrates the consequences of such an event (*Monteith et al., 2016*). However, while the requirement for tight regulation of TRIM21 signalling is intuitive the mechanism behind this has remained elusive. Our data suggest that it is achieved by a novel autoinhibitory mechanism in which the B-Box domain prevents ubiquitination by acting as an E2 mimic and the kinases IKKβ and TBK1 promote ubiquitination by phosphorylating the RING to relieve B-Box inhibition. Use of innate immune kinases intimately associates TRIM21 with the antiviral state not only at the level of gene expression as an ISG but also post translationally and in a similar manner to key immune adaptors MAVS, STING and TRIF (*Liu et al., 2015*).

## Materials and methods

### Cloning, expression and purification of recombinant proteins

Ube1, Ube2N, Ube2V2, Ube2D1, TRIM21$^{RING-Box}$ (residues 1–129) were produced as previously described (*Fletcher et al., 2015*). TRIM21$^{RING}$ (residues 1–85), TRIM5α$^{RING}$ (residues 1–88) and Ube2W were sub-cloned into pOP-TG for expression with a TEV cleavable GST tag. Point mutations were introduced using the site-directed mutagenesis protocol of Liu and Naismith (*Liu and Naismith, 2008*). Gibson cloning was used to remove the N-terminal GST tag and introduce a non-cleavable C-terminal STREP tag (*Gibson et al., 2009*). The miniTRIM21 construct was designed using the strategy of (*Wagner et al., 2016*). Residues 225–265 of a bacterial seryl-RNA synthetase, encoding an anti-parallel coiled-coil hairpin was inserted between two regions of the TRIM21 N-terminus, including residues 1–154 (RING-B-Box-coil) and residues 221–260 (coil). The resulting gene was cloned into pOP-TG for expression with a TEV cleavable GST tag at the N-terminus.

The *Escherichia coli* strain C41, was used for recombinant protein expression. Cells were grown in 2TY media supplemented with 100 µg/mL ampicillin at 37°C until an $OD_{600}$ of 0.7. The cells were induced with 1 mM IPTG and incubated over-night at 18°C. For expression of TRIM proteins, 1 mM IPTG was included upon induction. After harvesting, cells were resuspended in 50 mM Tris pH 8, 150 mM NaCl, 10 µM $ZnCl_2$, 1 mM DTT, 20% (vol/vol) Bugbuster (Novagen) and cOmplete protease inhibitors (Roche, Switzerland). The cell suspension was lysed by sonication, clarified by centrifugation at 18000 rpm for 40 min and passed over glutathione-sepharose resin (GE Healthcare). The fusion proteins were cleaved with TEV protease overnight at 4°C and were left with an N-terminal scar consisting of the tripeptide, Gly-Ser-His. Size exclusion chromatography was carried out on a High load 26/60 Superdex 75 prep grade column (GE Healthcare) as final purification step. All proteins were stored in 50 mM Tris pH 8, 150 mM NaCl, 1 mM DTT.

Isotopically labeled TRIM21$^{RING-M10E}$, TRIM21$^{RING-Box-M10E}$, TRIM21$^{RING-Box-M10E/M72E}$ proteins were produced in the Escherichia coli strain Rosetta 2(DE3). Cells were grown in K-MOPS minimal medium supplemented with 20 mM $^{15}NH_4Cl$ and/or 0.4% [$^{13}C$]-glucose, 4 mM $K_3PO_4$ pH 8.0, vitamin mix (0.1% Thiamine, 0.02% each d-Biotin, Choline chloride, Folic acid, Niacinamide, D-pantothenic acid, Pyridoxal, 0.002% Riboflavin) and 100 µg/ml pantothillin. Expression and purification conditions were as for unlabelled proteins, described above.

## Mass spectrometry

Polyacrylamide gel slices (1–2 mm) containing the proteins were prepared for mass spectrometric analysis using the Janus liquid handling system (PerkinElmer, UK). Briefly, the excised protein gel pieces were placed in a well of a 96-well microtitre plate and destained with 50% v/v acetonitrile and 50 mM ammonium bicarbonate, reduced with 10 mM DTT, and alkylated with 55 mM iodoacetamide. After alkylation, proteins were digested with 6 ng/µL trypsin (Promega, UK) overnight at 37°C. The resulting peptides were extracted in 2% v/v formic acid, 2% v/v acetonitrile. Digests were analysed by nano-scale capillary LC-MS/MS using an Ultimate U3000 HPLC (ThermoScientific Dionex, San Jose, USA) to deliver a flow of approximately 300 nL/min. A C18 Acclaim PepMap100 5 µm, 100 µm x 20 mm nanoViper (ThermoScientific Dionex, San Jose, USA), trapped the peptides prior to separation on a C18 Acclaim PepMap100 3 µm, 75 µm x 250 mm nanoViper (ThermoScientific Dionex, San Jose, USA). Peptides were eluted with a 60 min gradient of acetonitrile (2% to 80%). The analytical column outlet was directly interfaced via a nano-flow electrospray ionisation source, with a hybrid quadrupole orbitrap mass spectrometer (Q-Exactive Plus Orbitrap, ThermoScientific, San Jose, USA). Data dependent analysis was carried out, using a resolution of 30,000 for the full MS spectrum, followed by ten MS/MS spectra. MS spectra were collected over a m/z range of 300–2000. MS/MS scans were collected using a threshold energy of 27 for higher energy collisional dissociation (HCD). LC-MS/MS data were then searched against a protein database (UniProt KB) using the Mascot search engine programme (Matrix Science, UK) (*Perkins et al., 1999*). Database search parameters were set with a precursor tolerance of 10 ppm and a fragment ion mass tolerance of 0.8 Da. One missed enzyme cleavage was allowed and variable modifications for oxidized methionine, carbamidomethyl cysteine, pyroglutamic acid, phosphorylated serine, threonine and tyrosine, ubiquitylation at lysine and the N-terminus were included. MS/MS data were validated using the Scaffold programme (Proteome Software Inc., USA). All data were additionally interrogated manually.

## Analytical ultracentrifugation

Samples of TRIM21$^{RING}$, and mutants TRIM21$^{RING-M10E}$ and TRIM21$^{RING-M10E/M72E}$ at concentrations of 7.5 mg/ml were subjected to velocity sedimentation at 50,000 rpm at 20 °C in 50 mM Tris-HCl, pH 8.0, 150 mM NaCl, 1 mM DTT using 12 mm double sector cells in an An50Ti rotor using an Optima XL-I analytical ultracentrifuge (Beckmann). The sedimentation coefficient distribution function, c(s), was analyzed using the SEDFIT program, version 14.0 (*Schuck, 2003*). The partial-specific volumes (v-bar), solvent density and viscosity were calculated using Sednterp (Dr. Thomas Laue, University of New Hampshire).

## SEC-MALLS

Size-exclusion chromatography (SEC) was performed with inline multi-angle laser light scattering (MALLS) using a Wyatt HELEOS-II 18-angle photometer coupled to a Wyatt Optilab rEX differential refractometer (Wyatt Technology Corp). Samples of 100 µL were injected at 19.5 mg/mL and separated over a Superdex 75 10/300 GL (GE Healthcare) equilibrated in 50 mM Tris, pH 8, 150 mM NaCl, 1 mM DTT.

## In vitro ubiquitination assay

In vitro ubiquitination reactions were carried out in 50 mM Tris at pH 8, 2.5 mM MgCl2, 0.5 mM DTT with 0.2 mM Ub, 2 mM ATP, 1 µM Ube1, 0.5 µM Ube2N, 0.5 µM Ube2V2 1.5 or 1 µM TRIM21 RING-Box or RING protein respectively. The reaction was started upon incubation at 37°C for the time points indicated in the text. The reaction was stopped via the addition of LDS sample buffer at 4°C, followed by boiling at 90°C for 2 min. To observe the synthesis of TRIM21-anchored ubiquitin chains, 1.5 µM Ube2w was included in the reaction. The reactions were resolved by LDS-PAGE and TRIM21 or ubiquitin were detected by immunoblot using anti-TRIM21 [raised against human TRIM21$^{RING-Box}$ or TRIM21$^{RING}$], or anti-Ub-HRP (Santa Cruz, sc8017-HRP P4D1, 1:1,000).

## Single-turnover ubiquitin discharge assay

To prevent auto-ubiquitination, lysine 92 of Ube2N was replaced with arginine (*McKenna et al., 2001*). Ube2N$^{K92R}$ or Ube2D1 were loaded with ubiquitin by mixing 40 µM of the E2 with 1 µM Ube1, 0.37 mM Ub and 3 mM ATP in 50 mM HEPES pH 7.5, 150 mM NaCl, 20 mM MgCl$_2$, and

incubating the reaction at 37°C for 30 min. The reaction was transferred to ice and used immediately. To observe E3 mediated discharge of ubiquitin, 2 µM ubiquitin loaded E2 was mixed with 1.5 µM TRIM21$^{RING-Box}$ or TRIM21$^{RING}$ in 50 mM HEPES pH 7.5, 150 mM NaCl, 20 mM MgCl$_2$, 50 mM L-lysine. For discharge of ubiquitin from ubiquitin loaded Ube2N, 2.5 µM Ube2V2 was added to the reaction. Samples were taken at the time points indicated in the text and mixed immediately with LDS sample buffer at 4°C. The samples were boiled for exactly 20 s, resolved by LDS-PAGE and observed by immunoblot using anti-Ube2D (Boston Biochem, A-615, 1:1,000) or anti-Ube2N (Bio-Rad, AHP974, 1:1,000). For Ube2N immunoreactivity was observed by near-infrared detection (Odyssey, LI-COR).

## Co-translational phosphorylation

Gibson cloning was used to sub-clone the TRIM21$^{RING-Box}$ and MiniTRIM21 sequences into the pNHD1.3 plasmid (*Rogerson et al., 2015*) with the addition of a non-cleavable C-terminal Strep tag (GSWSHPQFEK). An S80TAG mutation introduced using the site-directed mutagenesis protocol of Liu and Naismith to create pNHD-T21RB(S80TAG) and pNHD-T21MT(S80TAG) (*Liu and Naismith, 2008*). Phosphorylated proteins were expressed in BL21 ΔserB(DE3) transformed with pKW2 EF-Sep and pNHD-T21RB(S80TAG) or pNHD-T21MT(S80TAG). Cells were grown at 37°C in TB media supplemented with 25 µg/mL chloramphenicol and 25 µg/ml tetracycline. At OD600 = 0.6, the cells were induced with 1 mM IPTG and 2 mM phosphoserine and incubated overnight at 18°C. Cells were harvested by centrifugation and resuspended in 100 mM Tris pH 8, 150 mM NaCl, 2 mM DTT, 10 µM ZnCl$_2$, 20% (vol/vol) Bugbuster (Novagen) and Complete protease inhibitors (Roche). The cells were lysed by sonication and centrifuged (18 000 rpm, 40 min, 4° C) to remove insoluble material. The soluble lysates were applied to 5 mL of StrepTactin Sepharose High Performance resin (GE Healthcare) that had been pre-equilibrated in wash buffer (100 mM Tris, pH 8, 150 mM NaCl, 1 mM DTT). The resin was washed in 10 column volumes of wash buffer and the protein was eluted in wash buffer with the addition of 2.5 mM D-desthiobiotin. Pooled protein fractions were subjected to size exclusion chromatography over a High load 26/60 Superdex 75 prep grade column (GE Healthcare) equilibrated in 50 mM Tris pH 8, 150 mM NaCl, 1 mM DTT. Incorporation of phosphoserine into the recombinant protein was confirmed by ESI-MS or LC-MS/MS and western blot.

## Dephosphorylation of recombinant phospho-proteins

Phosphorylated miniTRIM21 or the unmodified control were mixed with 400 U of Lambda protein phosphatase and buffered with 1 X NEBuffer (50 mM HEPES, 100 mM NaCl, 2 mM DTT, 0.01% Birj 35, pH 7.5 @ 25°C), supplemented with 1 mM MnCl$_2$. The reactions were incubated at 30°C for up to 30 min with samples taken at the times indicated in *Figure 4*. Dephosphorylation was confirmed by immunoblot using anti-TRIM21 and pS80 sera

## In vitro kinase assay

In vitro phosphorylation reactions were carried out in 10 µL reactions with 50 mM Tris pH 7.4, 10 mM MgCl$_2$, 0.5 mM DTT, 1 mM ATP, 430 ng IKKβ (Life Technologies), 100 or 400 ng TBK1 (Promega), 1 µM TRIM21$^{RING-Box}$ or LipoylTRIM21, incubated at 37°C for 2 hr, then quenched by addition of LDS sample buffer and boiling at 95°C for 5 min. Samples were resolved by LDS-PAGE and TRIM21 detected by immunoblot using anti-TRIM21 (clone D12, Santa Cruz) or pS80 sera at 1:10000 and 1:1000 dilutions, respectively.

## Cellular kinase assay

$10^6$ 293 T cells were transfected with 300 ng TRIM21-His (WT or S80A) and 3 µg empty vector or IKKβ (S177E/S181E). 48 hr later cells were washed, pelleted, denatured in 500 µL 6 M GuHCl, 0.1 M Na2HPO4/NaH2PO4 (pH 8), 10 mM Imidazole (pH 8) and samples rotated for 3 hr at room temperature with 30 µL equilibrated NiNTA agarose (Qiagen). The agarose matrix was washed twice with 500 µL lysis buffer, twice with 500 µL 3:1 wash buffer:lysis buffer, once with 500 µL wash buffer (25 mM Tris, 20 mM imidazole pH 6.8), resuspended in 2 × LDS sample buffer supplemented with 300 mM Imidazole to elute bound His-tagged proteins and heated for 5 min at 95°C before LDS-PAGE. For assaying IKKβ activity in the absence of NF-κB signaling, $10^6$ 293T were transfected with 500 ng pGL4.32[*luc2P*/NF-κB-RE/Hygro], 200 ng TRIM21-His and 2 µg empty vector or IKKβ. 6 hr later, cells

were treated with DMSO or 30 nM bortezomib for 16 hr. Cells were washed, resuspended in 500 μL cold PBS. 20 μL cells were mixed with 100 μL SteadyLite (Perkin Elmer) and luminescence measured (Pherastar). The remaining cells were pelleted, denatured and mixed with NiNTA agarose as described above. For TBK1 and MAVS transfection experiments, 200 ng TRIM21-His was transfected with 2 μg empty vector or TBK1 or MAVS. Cells were harvested at 24 hr post transfection.

## NF-κB reporter assay in 293T

293 T cells were seeded in 24 well plates 24 hr before transfection. Each well was transfected with, typically, 5 ng TRIM cDNA, with 10 ng pGL4.32[*luc2P*/NF-κB-RE/Hygro] (NF-κB response element-dependent firefly luciferase) and 5 ng pRL-TK (thymidine kinase promoter-dependent *Renilla* luciferase). 24 hr post transfection, cells were lysed in Passive Lysis Buffer and sequential firefly and renilla luminescence measured (BMG Pherastar plate reader), according to manufacturers instructions (Promega). Firefly luciferase luminescence was normalised to *Renillia* luciferase luminescence, and these values normalised to those of empty vector transfected cells.

## Generation of CRISPR knockout 293Ts

Single-guide RNA (sgRNA) against IKKβ (GCTGACCCACCCCAATGTGG) was incorporated into the Lenti CRISPR v2 plasmid (Addgene) and VSV-G pseudotyped lentiviral particles were generated by three-plasmid transfection of 293T with Fugene-6 (Promega), using 1 μg HIV-1 Gag-Pol expression plasmid, 1 μg VSV-G expression plasmid pMD2.G (GenScript), and 1.5 μg Lenti CRISPR v2 (IKKβ), or empty Lenti CRISPR v2 as a Cas9-encoding but sgRNA-negative control. $3 \times 10^5$ 293T were transduced with 50 μL 293T viral supernatant in the presence of 8 μg/mL polybrene (Santa Cruz), and selected with 2.5 μg/mL puromycin. Single cell clones were derived by limiting dilution. Loss of protein expression was confirmed by immunoblot. Generation of TRIM21 knockout 293Ts (T21KO) was achieved by electroporation (Neon) of Cas9/gRNA ribonucleoprotein complex (Cas9 RNP). The recombinant Cas9-2NLS-GFP protein was produced as described (*Jinek et al., 2012*). Synthetic tracrRNA and crRNA against *TRIM21* (ATGCTCACAGGCTCCACGAA) were obtained from Sigma-Aldrich. tracrRNA-crRNA complex was assembled by incubating at 20°C for 10 min. The RNA complex was combined with recombinant Cas9 protein at a molar ratio of 1:1.2 to form the Cas9 RNP complex following an incubation at 37°C for 10 min. Cas9 RNP against *TRIM21* was introduced into $8 \times 10^5$ 293 T cells using the Neon Transfection System (Invitrogen) with 2 pulses of 1400 V for 20 ms. 48 hr post electroporation the cells were cloned by fluorescence-activated cell sorting into 96-well plates (1 cell/well). Loss of protein expression was confirmed by immunoblot. 293T/T21KO/T21His were made by transducing T21KOs with T21His in pHR'.

## Mice

C57BL/6 wild-type mice were obtained from Jackson Laboratories. Thirteen week old-mice were used for immunisation experiment, which was conducted in accordance with the 19.b.6 moderate severity limit protocol and Home Office Animals (Scientific Procedures) Act (1986). All animal work was licensed under the UK Animals (Scientific Procedures) Act, 1986 and approved by the Medical Research Council Animal Welfare and Ethical Review Body. Mice were immunised subcutaneously (s. c) with 100 ug TRIM21 phophoS80-peptide ([67]RQLANMVNNLKEISQ[81]) conjugated to KLH in PBS mixed with complete Freund's adjuvant followed by 2 rounds of s.c boosting with 50 ug peptide mixed with incomplete Freund's adjuvant. Serum prepared from tail bleeds and cardiac blood was analysed by ELISA for binding to pS80-TRIM21 or TRIM21.

## Cells and viruses

WT and TRIM21-/- (K21) MEF cell lines were maintained in Dulbecco's Modified Eagle Medium (DMEM) supplemented with 10% fetal calf serum, penicillin at 100 U/ml and streptomycin at 100 μg/ml. MEF cells were obtained from WT or TRIM21 knockout C57/B6 mice and have previously been described and authenticated (*Guilliams et al., 2014*). Human adenovirus type five vector (ΔE1,ΔE3) expressing GFP (AdV) was purchased from ViraQuest. Human rhinovirus type 14 (HRV) was produced by infection of HeLa cells, and virus was purified by 2 rounds of CsCl centrifugation. 293T CRL-3216 cells were purchased from ATCC and authenticated by the supplier. All cells used are regularly tested and are mycoplasma free.

## Poly(I:C) stimulation of 293T

$10^7$ 293T/T21KO/T21His were transfected with 25 ug/mL poly(I:C) using Lipofectamine LTX (Life Technologies) and incubated for 5 or 10 min, or left untransfected. Cells were then washed, pelleted and denatured in 500 μL 6 M GuHCl, 0.1 M Na2HPO4/NaH2PO4 (pH 8), 10 mM Imidazole (pH 8) and pull-downs performed as in the 'cellular kinase assay' described above.

## Adenovirus stimulation of TRIM21 phosphorylation

$2.5 \times 10^{11}$ pts/mL AdV vector was mixed with 10 μg/mL monoclonal anti-hexon 9C12 and incubated for 1 hr at room temperature. $10^7$ 293T/T21KO/T21His were transduced with virus/antibody mixture, or treated with equal volume of PBS, and incubated at 37C for 7 hr. Cells were then washed, pelleted and denatured in 500 μL 6 M GuHCl, 0.1 M Na2HPO4/NaH2PO4 (pH 8), 10 mM Imidazole (pH 8) and pull-downs performed as in the 'cellular kinase assay' described above.

## Lentiviral transduction

Human TRIM21 WT and mutants S80E, S80A were PCR amplified, cloned into lentiviral vector pHR' (kind gift of Dr. Adrian Thrasher) using NotI and SalI restriction sites and sequence verified. Lentiviral particles were generated by co-transfection of 10 cm$^2$ dishes of 293Ts with 1 ug pCRV GagPol, 1 ug pMD2G VSVg and 1.5 ug TRIM21 pHR' using Fugene-6 (Promega). K21 MEF cells were infected with lentivirus in the presence of 8 μg/mL polybrene and stably transduced cells were selected using puromycin at 2 μg/ml. For the generation of stably expressed TRIM21-His, we cloned TRIM21-His into pHR', prepared lentivectors as above, transduced 293T TRIM21KO cells in the presence of 8 μg/mL polybrene, and selected cells with 2.5 μg/ml puromycin.

## Cytokine analysis by qPCR

Cells were plated at $1 \times 10^4$ per well in 96 well plates. After 24 hr, viruses were mixed 1:1 with antibody and incubated for 30 mins at room temperature prior to infection. AdV was used at $1.25 \times 10^9$ pts per well mixed with 10 mg/ml human serum IgG (Sanquin). HRV was used at $2.5 \times 10^2$ TCID$_{50}$ units per well mixed with 5 mg/ml human serum IgG (Sanquin). Cells were incubated at 37° C for 3.5 hr, before washing with PBS and preparation of cDNA using Cells to CT Kit (Ambion). Gene expression was monitored by TaqMan Gene Expression Assays (Applied Biosystems) on a StepOnePlus Real Time PCR System (Life Technologies). Gene expression assays were: mouse β-actin (4352933E), CxCL10 (Mm00445235_m1), TNF (Mm00443260_g1), IL-6 (Mm00446190_m1), IFIT1 (Mm00515153_m1). Relative expression was quantified using the $2^{-\Delta\Delta Ct}$ method.

## Virus neutralization assay

Cells were seeded into 24-well plates at $5 \times 10^4$ cells/well and allowed to adhere overnight. Virus was diluted to $1.5 \times 10^8$ pts in 6 μl in phosphate-buffered saline (PBS), mixed with an equal volume of 9C12 antibody variant at the stated concentration, and incubated for 30 min to allow binding to reach equilibrium. NAb-virus mixes were diluted 100-fold into cell culture medium and added to cells. GFP-positive cells were analyzed by flow cytometry after 24 hr of incubation at 37°C. Levels of infection in the absence of NAb were in the range of 50% to 70%. Relative infection was calculated by dividing the percent GFP-positive cells in the presence of antibody to that of a PBS-treated control infection.

## Crystallography

TRIM21$^{RING-Box}$ protein at 10 mg/mL was buffer exchanged into 50 mM Tris pH 8, 150 mM NaCl, 1 mM DTT and subjected to sparse matrix screening in sitting drops at 17°C. Crystals grew reproducibly in 10 % w/v PEG 4000, 20% v/v glycerol, 0.03 M diethylene glycol, 0.03 M trithylene glycol, 0.03 M tetraethylene glycol, 0.03 M pentaethylene glycol, 0.055 M MES, 0.045 M Imidazole, pH 6.5. The crystals were cryoprotected in 20% Ethylene Glycol and flash frozen in liquid Nitrogen. Diffraction experiments were performed at the Diamond light source on the I04-1 beamline, which was equipped with a PILATUS 2M detector. A 1.95 Å resolution data set was collected at a wavelength of 0.92 Å. TRIM21$^{RING-Box}$ protein contained four zinc atoms per monomer, allowing for the collection of SAD data. Data indexing and scaling were performed with MOSFLM and AIMLESS respectively (*Winn et al., 2011*). Experimental phasing, density modification and initial model

building was performed using *AutoSol* from the *PHENIX* suite (*Adams et al., 2010*). Further model building was performed in Coot (*Emsley and Cowtan, 2004*). REFMAC 5.7 was used for model refinement, with initial rounds of restrained refinement including phase information from Hendrickson-Lattman coefficients (*Murshudov et al., 1997*). In later rounds, Translation/liberation/screw (TLS) refinement was introduced with individual polypeptide chains defined as TLS groups. MolProbity, which was used for model validation, gave ramachandran statistics of: 95.2% of residues were found in favoured regions and 100% in allowed regions (*Chen et al., 2010*). The structure factors and coordinates can be found in the Protein Data Bank, with the accession number 5OLM.

## NMR spectroscopy

$^{13}C/^{15}N$ or $^{15}N$ isotopically labelled TRIM21 proteins were buffer exchanged into 50 mM $[^2H_{11}]$-tris, pH 7.0, 150 mM NaCl, 1 mM $[^2H_{10}]$-DTT, $H_2O/^2H_2O$ 95:5. Spectra were recorded at 25°C on Bruker DRX 500 and Avance I 600 spectrometers (Bruker BioSpin GmbH), each equipped with a ($^1H/^{15}N/^{13}C$) 5 mm cryoprobe, and were processed using the program Topspin (Bruker BioSpin GmbH) and analysed using the program Sparky (Goddard). Assignments were made using $^{13}C/^{15}N$ labelled protein and a standard suite of NMR experiments ([$^{15}N,^1H$] HSQC, [$^{13}C,^1H$] HSQC, HNCA, HNCOCA, CBCANH, CBCA(CO)NH, HBHANH and HBHA(CO)NH), allowing 93% of TRIM21$^{RING-M10E}$, 97% of TRIM21$^{RING-Box-M10E}$ and 95% of TRIM21$^{RING-Box-M10E/S80E}$ amide resonances of non-proline residues to be assigned. Chemical shift perturbations (CSPs) were measured using samples of $^{15}N$-labelled 90 μM TRIM21$^{RING-M10E}$, 200 μM TRIM21$^{RING-Box-M10E}$ or 200 μM TRIM21$^{RING-Box-M10E/S80E}$, to which were added either 0.5 or 1 molar equivalents of unlabeled Ube2N; the reported CSP values are all for 1:1 complexes and were calculated according to the formula $((\Delta\delta(^1H))^2 + (\Delta\delta(^{15}N)/5)^2)^{1/2}$.

## Relaxation analysis

$^{15}N\{^1H\}$ heteronuclear NOE ratios for the amide signals of TRIM21$^{RING-Box-M10E}$ and TRIM21$^{RING-Box-M10E/S80E}$ were measured using 200 μM $^{15}N$ labelled samples and a pre-saturation time of 7 s, essentially as described by (*Skelton et al., 1992*).

## Solvent PRE

Solvent paramagnetic relaxation enhancement (PRE) data were measured by the addition of the soluble contrast agent, Gd-DTPA. A 100 mM stock solution of Gd-DTPA was made in 50 mM $[^2H_{11}]$-tris and the pH adjusted to 7.0; this solution was added to a final concentration of 4 mM to samples of 200 μM TRIM21$^{RING-Box-M10E}$ or 200 μM TRIM21$^{RING-Box-M10E/S80E}$. Peak intensities for each amide signal in the [$^{15}N,^1H$] HSQC spectra were recorded before ($I^{REF}$) and after ($I^{PRE}$) addition of the contrast agent; for *Figures 4H* and *6*, the reported PRE ratio is calculated as ($[I^{PRE}/I^{REF}]$WT)/($[I^{PRE}/I^{REF}]$S80E).

## Acknowledgements

This work was funded by the Medical Research Council (UK; U105181010 and U105178934) and a Wellcome Trust Investigator Award. We thank Dr. Jonathan Pruneda for assistance with E2 discharge assays. CFD was supported by an NHMRC Early Career Fellowship (APP1110116).

## Additional information

### Funding

| Funder | Grant reference number | Author |
| --- | --- | --- |
| National Health and Medical Research Council | Early Career Fellowship APP1110116 | Claire Dickson |
| Medical Research Council | U105181010 | David Neuhaus Leo C James |
| Medical Research Council | U105178934 | David Neuhaus |
| Wellcome Trust | 200594/Z/16/Z | Leo C James |

The funders had no role in study design, data collection and interpretation, or the decision to submit the work for publication.

## Author contributions
Claire Dickson, Data curation, Formal analysis, Investigation, Writing—review and editing, Conceived the project and wrote the manuscript, Performed X-ray crystallography and in vitro ubiquitination assays; Adam J Fletcher, Formal analysis, Investigation, Writing—review and editing, Conceived the project and wrote the manuscript, Carried out cellular experiments and kinase assays; Marina Vaysburd, Investigation, Carried out virus-induced cytokine infections; Ji-Chun Yang, Investigation, Performed NMR experiments; Donna L Mallery, Investigation, Assisted with ubiquitination assays; Jingwei Zeng, Investigation, Assisted with cell experiments, Provided new reagents; Christopher M Johnson, Investigation, Performed multi-angle light scattering; Stephen H McLaughlin, Investigation, Performed AUC; Mark Skehel, Supervision; Sarah Maslen, Investigation; James Cruickshank, Supervision, Investigation; Nicolas Huguenin-Dezot, Methodology; Jason W Chin, Supervision, Methodology; David Neuhaus, Data curation, Supervision, Analysed and supervised NMR; Leo C James, Conceptualization, Formal analysis, Supervision, Writing—original draft, Writing—review and editing, Conceived the project and wrote the manuscript

## Author ORCIDs
Stephen H McLaughlin https://orcid.org/0000-0001-9135-6253
Leo C James https://orcid.org/0000-0003-2131-0334

## Decision letter and Author response
Decision letter https://doi.org/10.7554/eLife.32660.021
Author response https://doi.org/10.7554/eLife.32660.022

# Additional files

## Supplementary files
• Supplementary file 1. Data collection and refinement statistics
DOI: https://doi.org/10.7554/eLife.32660.017
• Supplementary file 2. Solvent PRE raw data
DOI: https://doi.org/10.7554/eLife.32660.018
• Transparent reporting form
DOI: https://doi.org/10.7554/eLife.32660.019

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
