## [Decision Letter]

Thank you for submitting your article "Intracellular antibody signalling is regulated by IKK phosphorylation of the Fc receptor TRIM21" for consideration by *eLife*. Your article has been favorably evaluated by Wenhui (Senior Editor) and three reviewers, one of whom is a member of our Board of Reviewing Editors.

The reviewers have discussed the reviews with one another and the Reviewing Editor has drafted this decision to help you prepare a revised submission.

Summary:

TRIM21 is a unique cytosolic IgG receptor that detects antibody-coated pathogens in the cytosol. Detection of pathogen-antibody complex by TRIM21 drives two distinct events; 1) TRIM21 drives degradation of the pathogen-antibody complex by recruiting the VCP/p97 and proteasome, 2) TRIM21 activates innate immune signaling by generating unanchored K63-linked chains.

Here, the authors revealed one of the regulatory mechanisms of TRIM21 ubiquitin ligase activity that leads to the generation of K63 chains. Ligase activity of TRIM21 is constitutively auto-inhibited by its B-box domain. The authors showed that B-box binds to its RING domain and inhibits E2 recruitment by using crystal structure and NMR analysis. The authors also show that phosphorylation at S80 in the RING domain by IKK disrupts the B-box-RING binding and releases B-box inhibition to activate its ubiquitin ligase activity.

This finding is important for understanding the regulatory mechanisms of TRIM family ubiquitin ligase, and this study is generally and technically well performed and includes appropriate controls. However, two issues should be addressed:

1) In regard to TBK1, whether TBK1 can also phosphorylate TRIM21 or not;

2) Whether physiological stimulation can induce phosphorylation of TRIM21 or not.

Essential revisions:

In Figure 5, the authors detected the phosphorylation of S80 by IKK overexpression. It would be valuable to learn whether S80 is or is not phosphorylated in physiological conditions. It is well known that IKK is activated by various stimuli such as TNF-, IL-1, LPS. When IKK is activated by such a cytokine, can phosphorylation of S80 be detected? If so, is TRIM21 involved in the enhancement of cytokine stimulation? Moreover, S in the *p*LxIS motif is also phosphorylated by TBK1 (Liu et al., 2015). It is of great interest to know whether or not S80 is also phosphorylated by TBK1 in overexpression vs. more physiological settings.

---

## [Author Response]

[…] This finding is important for understanding the regulatory mechanisms of TRIM family ubiquitin ligase, and this study is generally and technically well performed and includes appropriate controls. However, two issues should be addressed:1) In regard to TBK1, whether TBK1 can also phosphorylate TRIM21 or not;2) Whether physiological stimulation can induce phosphorylation of TRIM21 or not.Essential revisions:In Figure 5, the authors detected the phosphorylation of S80 by IKK overexpression. It would be valuable to learn whether S80 is or is not phosphorylated in physiological conditions. It is well known that IKK is activated by various stimuli such as TNF-, IL-1, LPS. When IKK is activated by such a cytokine, can phosphorylation of S80 be detected? If so, is TRIM21 involved in the enhancement of cytokine stimulation? Moreover, S in the pLxIS motif is also phosphorylated by TBK1 (Liu et al., 2015). It is of great interest to know whether or not S80 is also phosphorylated by TBK1 in overexpression vs. more physiological settings.

To address whether TRIM21 is phosphorylated under physiological conditions, we stimulated cells in three different ways: by stimulating the pathway immediately upstream of IKK and TBK1 by expressing MAVS, by challenging cells with the MDA5/RIG-I ligand poly(I:C) and by infecting cells with adenovirus. We chose poly(I:C) as our stimulating ligand rather than LPS because 293T cells lack the LPS receptor TLR4. In all three separate experiments, we observed an increase in TRIM21 phosphorylation (Figure 6C-E). TRIM21 is not constitutively required to enhance cytokine stimulation during immune stimulation, as we have shown in Watkinson et al., 2015, but preventing S80 phosphorylation does inhibit TNF induction by TRIM21 during infection in the presence of antibody (present manuscript, Figure 8C). In addition to this data, we have performed experiments to show that TBK1 phosphorylates TRIM21 both in vitro and in cells (Figure 6A-B). We are extremely grateful for the suggested revisions, as showing TRIM21 phosphorylation by different kinds of immune stimuli and by both IKK and TBK1 has significantly bolstered the significance of our findings.